# Platelet-Derived Growth Factor Induces SASP-Associated Gene Expression in Human Multipotent Mesenchymal Stromal Cells but Does Not Promote Cell Senescence

**DOI:** 10.3390/biomedicines9101290

**Published:** 2021-09-22

**Authors:** Olga Grigorieva, Mikhail Arbatskiy, Ekaterina Novoseletskaya, Uliana Dyachkova, Alexander Ishkin, Natalia Kalinina, Anastasia Efimenko

**Affiliations:** 1Institute for Regenerative Medicine, Medical Research and Education Center, Lomonosov Moscow State University, Lomonosovsky Ave., 27/10, 119192 Moscow, Russia; go.grigorievaolga@gmail.com (O.G.); kuznecova2793@mail.ru (E.N.); 2Faculty of Medicine, Lomonosov Moscow State University, Lomonosovsky Ave., 27/1, 119991 Moscow, Russia; algenubi81@mail.ru (M.A.); dyachkovauliana@gmail.com (U.D.); n_i_kalinina@mail.ru (N.K.); 3Discovery Science, Clarivate Analytics, Boston, MA 02210, USA; don.shikin@gmail.com

**Keywords:** cellular senescence, transcriptome, multipotent mesenchymal stromal cells, platelet-derived growth factor

## Abstract

Activation of multipotent mesenchymal stromal cells (MSCs) is a central part of tissue response to damage. Platelet-derived growth factor (PDGF-BB), which is abundantly released in the damaged area, potently stimulates the proliferation and migration of MSCs. Recent evidence indicates that tissue injury is associated with the accumulation of senescent cells, including ones of MSC origin. Therefore, we hypothesized that PDGF-BB induces MSC senescence. To evaluate mechanisms of early activation of MSCs by PDGF-BB, we performed transcriptome profiling of human MSCs isolated from adipose tissue after exposure to PDGF-BB for 12 and 24 h. We demonstrated that PDGF-BB induced the expression of several genes encoding the components of senescence-associated secretory phenotype (SASP) in MSCs such as plasminogen activator inhibitor-1 (PAI-1), urokinase-type plasminogen activator and its receptor (uPA and uPAR), and some matrix metalloproteases. However, further experimental validation of transcriptomic data clearly indicated that PDGF-BB exerted mitogenic and pro-migratory effects on MSCs, and augmented their pro-angiogenic activity, but did not significantly stimulate MSC senescence.

## 1. Introduction

Recently, tissue injury was associated with the accumulation of senescent cells, including those originating from multipotent mesenchymal stem/stromal cells (MSCs) [1]. Numerous studies have implicated MSCs in tissue regeneration upon damage, with paracrine activity being a central mechanism of their action [2,3,4,5]. Signals from damaged tissues could both activate MSCs and cause the acquisition of a senescent phenotype characterized by cell cycle arrest with p16INK4a, p21Waf1/Cip1, and p53 expression activation, resistance to apoptosis, and specific senescence-associated secretory phenotype (SASP) activated by transcription factors NF-κB and C/EBPβ [6,7,8]. SASP is characterized by a shift towards pro-inflammatory, mitogenic cytokines as well as changes in the extracellular matrix composition. MSC senescence causes the crucial impairment of their regenerative capacity and regulatory functions.

Upon damage, MSCs respond to a multitude of factors, including abundantly presented platelet-derived growth factor (PDGF). These cells express PDGFRβ, which has the highest affinity for PDGF-BB dimer [9]. PDGF can activate both the migration and proliferation of stromal cells [7,10], therefore stimulating tissue regeneration. However, PDGF-mediated activation of stromal cells could also contribute to fibrogenesis [11].

Since proliferation and senescence are closely related, we hypothesized that PDGF-BB could simultaneously cause MSCs’ senescence. To test this hypothesis, we performed transcriptomic profiling of human adipose-derived MSCs after treatment with PDGF-BB by RNA sequencing. Acquisition of the senescent phenotype by MSCs treated with PDGF-BB was evaluated by beta-galactosidase and p21 expression, as well as secretion of SASP components (IL-6, MCP-1, PAI-1). We confirmed mitogenic and pro-migratory effects of PDGF-BB on MSCs but did not observe the significant stimulation of MSC senescence in these conditions.

## 2. Materials and Methods

### 2.1. Human MSC Isolation and Culture

Human adipose-derived MSCs were isolated from subcutaneous adipose tissue obtained from 4 healthy donors during abdominal surgery. All donors, or if donors were under 18, a parent and/or legal guardian, gave their informed consent and the local ethics committee of the Medical Research and Education Center of Lomonosov Moscow State University (IRB00010587, Moscow, Russia) approved the study protocol (#4, 4 June 2018).

Subcutaneous adipose tissue samples (0.5–5 mL) harvested during surgery were homogenized and digested in collagenase I (200 U/mL, Worthington Biochemical, Lakewood, NJ, USA) and dispase (40 U/mL, Sigma, St. Louis, MO, USA) solution under agitation for 30–40 min at 37 °C. Then, tissue was centrifuged at 200× *g* for 10 min and the supernatant discarded. The pellet containing MSCs was lysed to destroy erythrocytes, filtered through a sieve (BD Falcon Cell Strainer, pore size 100 mkm), and centrifuged at 200× *g* for 10 min. The final pellet was resuspended in culture medium [12]. Cells were cultured in AdvanceSTEM™ Mesenchymal Stem Cell Media containing 10% AdvanceSTEM™ Supplement (both HyClone, Logan, UT, USA), 100 U/mL penicillin, and 100 µg/mL streptomycin (HyClone) at 37 °C in a 5% CO_2_ incubator. Cells were passaged at 70% confluency using HyQTase solution (HyClone). For the experiments, MSCs of the third or fourth passages were seeded at a density of 7 × 10^3^ cells/cm^2^ on TCP culture plates (Corning) a day before the experiment. At these passages, phenotypic characterization and functional evaluation of mesenchymal properties were performed as described below. MSCs were washed thoroughly with HBSS (Hanks’ balanced salt solution, HyClone) to remove any residues of AdvanceSTEM™ Supplement. Then, HBSS was replaced with serum-free AdvanceSTEM™ Mesenchymal Stem Cell Media, and MSCs were cultivated overnight for 18 h. For assessment of transcriptome content, MSCs were cultured under standard conditions (5% CO_2_ and 21% O_2_) at 37 °C for 12 or 24 h with or without 10 ng/mL PDGF-BB. At the end of the incubation, cells were washed with HBSS and were lysed in the RLT buffer (Qiagen, Venlo, Netherlands) for further isolation of the total RNA. MSCs quantity and viability were assessed at the end of experiment using Countess II Automated Cell Counter (Invitrogen, Waltham, MA, USA).

### 2.2. MSC Immunophenotyping

MSC immunophenotype was analyzed using flow cytometry. After medium harvesting, cells were detached from culture dishes using EDTA solution and stained with antibodies against CD14 (eBioscience, San Diego, CA, USA, 14-0141-81), CD34 (BD Pharmingen, San Diego, CA, USA, 555824), CD45 (BD Pharmingen, 340953), CD73 (BD Pharmingen, 550257), CD90 (BD Pharmingen, 555597) and CD105 (BD Pharmingen, 560819), as described elsewhere. IgGs of appropriate isotype were used as a negative control. Stained cells were analyzed using MoFlo cell sorter and Summit V 4.0. software (DakoCytomation, Glostrup, Denmark).

### 2.3. MSC Differentiation Assays

MSC abilities to differentiate into osteogenic, adipogenic, and chondrogenic directions were tested in vitro using standard differentiation and analysis protocols described elsewhere [13]. Briefly, osteogenic differentiation was induced by plating 6 × 10^4^ of MSCs into 24-well plates and incubating them in growth medium containing 10^−8^ M dexamethasone, 10 mM β-glycerol-2-phosphate, and 0.2 mM 2-phospho-l-ascorbic acid (Invitrogen) for 3 weeks. Differentiation efficiency was analyzed using Alizarin Red S staining. Adipogenic differentiation was induced by 3 cycles of consecutive incubation for 6 days in the growth medium containing 10^−6^ M dexamethasone, 10 µM insulin, 200 µM indomethacin, and 0.5 mM 3-isobutyl-1-methylxantine (Invitrogen), followed by 3 days of incubation in growth medium containing 10 µM insulin. Accumulation of intracellular lipids was assessed by Oil Red O staining. Chondrogenic differentiation was induced by incubation of pelleted MSCs in AdvanceSTEM Chondrogenic Differentiation Medium (HyClone) for 21 days as described in [14] and stained with Toluidine Blue.

### 2.4. HUVEC Isolation and Culture

Human umbilical cord endothelial cells (HUVEC) were obtained from the cell culture Depository of Live Systems (Lomonosov Moscow State University), ID of the collection MSU_HUVEC. HUVEC were cultured in EGM™ Media containing 2% fetal bovine serum (FBS) and recombinant growth factors (CC-4176, Lonza) at 37 °C in a 5% CO_2_ incubator. Cells were passaged at 70% confluency using HyQTase solution (HyClone). For migration assay, cells were used between passage 3 and 6.

### 2.5. RNA Isolation and Transcriptome Analysis

Total cellular RNA was isolated from cultured MSCs using an RNeasy Kit (Qiagen) according to manufacturer instructions. Five hundred nanograms of total RNA was used for transcriptome analysis.

### 2.6. Bead Array Hybridization

Expression profiling was carried out using Illumina HumanWG-6 BeadChip ((Illumina, San Diego, CA, USA) microarrays. The RNA sample quality was controlled using a BioAnalyser and the RNA 6000 Nano Kit (both Agilent, Santa Clara, CA, USA). PolyA RNA was purified with the Dynabeads^®®^ mRNA Purification Kit (Ambion, Austin, TX, USA). An Illumina library was made from polyA RNA with NEBNext^®®^ Ultra™ II RNA Library Prep (New England Biolabs, Ipswich, MA, USA), according to the manufacturer’s instructions. Concentrations of nucleic acids in the obtained libraries were analyzed by the Qubit dsDNA HS Assay Kit using Qbit 2.0 equipment (Thermo Fisher Scientific, Waltham, MA, USA). The distribution of library fragment lengths was assessed using the Agilent High-Sensitivity DNA Kit (Agilent, USA). cDNA was hybridized to HumanWG-6 BeadChip (Illumina^®^, San Diego, CA, USA) according to manufacturer instructions.

### 2.7. Data Preprocessing

The Illumina BeadStudio [15] output file was processed using the lumi package available in Bioconductor software collection [16]. The processing included the following steps: quality control of raw data; background subtraction; and variance stabilization using algorithm VST, developed for Illumina data. Values after this processing step should be treated as log-transformed.

Quantile normalization was performed. First, pairwise correlations of intensities were consistently high (>0.95) for the samples. Additionally, we excluded all probes that had detection *p*-value > 0.05 in all 3 samples. A total of 4 columns (samples) and 11,073 rows (probes) corresponded to 9659 non-redundant genes. Fold changes in gene expression relative to the expression in the control sample were calculated.

### 2.8. Enrichment Analysis

Enrichment analysis was conducted with the GOstats package [17] of Bioconductor. This analysis aims to identify biological processes (in our case, the “BP” branch of the Gene Ontology [18] biological processes tree) which are significantly associated with observed expression changes.

The significance of enrichment is defined by *p*-values of hypergeometric distribution:(1)$$p-value=\frac{R!n!(N-R)!(N-n)!}{N!}\sum_{i=\max(r,R+n-N)}^{\min(n,R)}\frac{1}{i!\left(R-i\right)!(n-i)!(N-R-n+i)i},$$
where

*N*—number of genes in the whole data set;

*R*—number of genes in a gene list of interest;

*n*—number of genes associated with a particular process from the ontology;

*r*—number of gene from input list intersection with genes from a particular category.

### 2.9. Data Records

CHIP files associated with the samples analyzed in this study were deposited in the Gene Expression Omnibus with the accession number GSE152578.

### 2.10. Immunohistochemistry

Human adipose-derived MSCs from 4 healthy donors were seeded at a density of 10^4^ per mL. Cells were used between passage 3 and 4. PDGF-BB was added at concentration 10 ng/mL for 24 h after 18 h of serum deprivation. MSCs were fixed with 4% paraformaldehyde solution (Panreac, Barselona, Spain) at room temperature for 10 min and incubated with 0.2% triton ×100 (Sigma) solution at RT for 10 min. Further, MSCs were incubated for 1 h in 1% bovine serum albumin (BSA, Sigma) and 10% normal goat serum (Abcam, Cambridge, UK) solution at room temperature to block the non-specific interaction of antibodies. Subsequently, the samples were incubated with primary polyclonal rabbit antibody for p21Waf1/Cip1 (Cell Signaling, 2947S) or rabbit polyclonal IgG (Biolegend, San Diego, CA, USA, 910801) in 1% BSA solution at +4° overnight. Then, samples were incubated with fluorescence-labeled goat anti-rabbit secondary antibodies (A11034, Invitrogen) at room temperature for 1 h. Cell nuclei were labeled with DAPI (DAKO). Samples were analyzed with a Leica DM6000B fluorescent microscope equipped with a Leica DFC 360FX camera (Leica Microsystems GmbH, Wetzlar, Germany). The percentage of p21-positive MSCs was evaluated.

### 2.11. β-Galactosidase Staining

Evaluation of β-galactosidase activity proceeded with the Senescence β-Galactosidase Staining Kit (Cell Signaling, Danvers, MA, USA, 9860) according to the manufacturer’s recommendations.

### 2.12. ELISA Assay

MSCs from 4 healthy donors were used between passages 3–4. Cells were seeded at density of 10^4^ per ml and allowed to attach to TCP plastic. After 18 h of serum deprivation, PDGF-BB were added at a concentration of 10 ng/mL for 24 h. Afterwards, MSCs were washed thoroughly with HBSS (HyClone), then HBSS was replaced with DMEM low glucose (Gibco, Waltham, MA, USA) with 1% BSA, and MSCs were cultivated for another 72 h. Condition medium was collected and centrifuged at 300× *g* for 10 min to remove cellular debris. Supernatant was transferred to a clear tube with 1% protease inhibitor cocktail ×100 (Sigma). The concentrations of IL-6, MCP-1 (CCL-2), PAI-1, HGF, and VEGF in the samples of conditioned medium were evaluated with ELISA kit (R&D Systems, Minneapolis, MN, USA) according to the manufacturer’s recommendations and normalized to cell number.

### 2.13. Migration Assay

HUVEC migration assay was performed in automated RTCA xCELLigence™ system (ACEA Biosciences, San Diego, CA, USA) according to the manufacturer’s recommendations. Briefly, HUVEC media was changed to basal media EBM (Lonza, Basel, Switzerland) with 0.5% bovine serum albumin (BSA, Imtek, Russia) to harvest serum growth factors 4–6 h before experiment. The lower chamber of the CIM plate was filled with EBM with 0.5% BSA as a negative control, EGM2, containing 2% of fetal bovine serum (FBS) (Lonza), and conditioned media from MSC (CM MSC) or conditioned media from MSC pretreated with 10 ng/mL PDGF-BB for 24 h. As a control for the possible chemotactic activity of PDGF itself, it was added to one of the chambers at a concentration of 10 ng/mL in EBM 0.5% BSA. A total of 20,000 HUVEC were placed in the upper chamber. After placing the CIM plate into the RTCA xCELLigence system, bioimpedance on each well was fixed every 15 min. Data were processed with RTCA Software Pro Version 2.3.2 (ACEA Biosciences).

### 2.14. Statistical Analysis

We performed at least three independent experiments for each measurement. Statistical data processing was performed using the program GraphPad Prism 8.0 (GraphPad Software, San Diego, CA, USA). Means ± standard deviation (SD) were used to present quantitative data. The comparison of independent groups was performed by Mann–Whitney test. Differences were considered statistically significant at the significance level of *p* < 0.05.

## 3. Results

### 3.1. PDGF-BB Upregulates Genes Associated with Proliferation and Cell Cycle and Downregulates Immune-Response-Related Pathways in MSCs

MSCs isolated from the adipose tissue of healthy donors comply with recommendations of the International Society for Cellular Therapy [19]. These cells possessed CD73^+^/CD90^+^/CD105^+^/CD45^−^/CD34^−^/CD14^−^/CD19^−^ immunophenotypes and differentiated into osteogenic, chondrogenic, and adipogenic lineages upon being cultured in the induction medium (Figure 1). Most cells were also positive for PDGFRβ (87–95%).

To evaluate the effect of PDGF-BB on global changes in gene expression in MSCs, we performed expression profiling of cells treated by PDGF-BB for 12 or 24 h using RNA sequencing. Differential expression analysis revealed numerous expression changes induced by PDGF-BB treatment (152 and 127 genes were upregulated at least 2-fold after 12 and 24 h exposure to PDGF-BB, correspondingly; 173 and 163 genes were downregulated at least 2-fold after 12 and 24 h exposure to PDGF-BB) (Appendix A).

Functional enrichment analysis was performed using the clusterProfiler Bioconductor package [20]. As a result, each differentially expressed gene list was associated with quantitatively ranked lists of Gene Ontology biological processes summarizing its effects at a systems level.

Main biological processes associated with differently expressed genes are presented in Figure 2 and Figure 3. Upregulated gene lists for experimental samples are clearly associated with proliferation and cell cycle. Among the genes of these groups, the expression was upregulated more than three times for CCNA2 (cyclin A2), CCNB2 (cyclin B2), CDC20 (cell division cycle 20), CDCA5 (cell division cycle-associated 5), KIAA0101 (PCNA-associated factor), TOP2A (topoisomerase (DNA) II alpha), TYMS (thymidylate synthetase), AURKA (aurora kinase A), AURKB (aurora kinase B), and NUSAP1 (nucleolar-associated protein 1). Expression of these genes indicates that cells actively proliferate after PDGF-BB treatment. This correlates with decreased accumulation of CDK inhibitors, such as p16INK4a, p21Waf1/Cip1, and p53, the markers of senescent cells. Additionally, we found a significant increase in the expression of genes regulating MSC migration, such as PLAU (urokinase-type plasminogen activator, urokinase), PLAUR (urokinase-type plasminogen activator receptor, uPAR), MMP-1 (matrix metalloprotease-1), and SERPINE1 (plasminogen activator inhibitor-1, PAI-1) which were also related to SASP.

Downregulated genes were associated with immune response-related pathways and processes, most prominently the gamma-interferon response. A significant decrease in expression was observed for chemokine genes, including CCL8 (C-C motif chemokine ligand 8, IL-8), CXCL10 (C-X-C motif chemokine ligand 10), CCL2 (C-C motif chemokine ligand, MCP-1), and CXCL12 (C-X-C motif chemokine ligand). In addition, there was a strong decrease in IL6 gene expression. Among genes downregulated in MSCs after PDGF-BB treatment, we also found THBS2 (thrombospondine-2). Figure 3 illustrates similarities in enriched biological processes between 12 and 24 h treatments.

### 3.2. PDGF-BB-Treated MSCs Did Not Acquire Senescence Biomarkers

Senescent cells are characterized by accumulation of cell cycle inhibitors, such as p16INK4a, p21Waf1/Cip1, and p53 [21]. We evaluated the number of MSCs with p21-positive nuclei with and without PDGF-BB treatment for 24 h. We did not find significant differences between treated vs. non-treated MSCs (Figure 4A). Furthermore, we evaluated the accumulation of senescent cells by the activity of β-galactosidase. There was no significant elevation of β-gal activity in PDGF-BB-treated MSCs (Figure 4B), which indicated that PDGF-BB did not induce cellular senescence.

Senescent-associated secretory phenotype (SASP) is characterized by the elevated production of cytokines and interleukins which mediates the proinflammatory microenvironment of senescent cells. Among possible SASP factors in the conditioned medium of MSCs, we measured such crucial factors as IL-6, MCP-1 (CCL2), and PAI-1 by ELISA. Obtained protein concentrations were normalized by cell number. In the case of IL-6, we found a significant decrease in the conditioned medium of PDGF-BB-treated MSCs compared with the non-treated control. Concerning MCP-1 and PAI-1, there were no significant changes in levels of secreted proteins between treated vs. non-treated cells (Figure 4C).

### 3.3. PDGF-BB Stimulated MSC Pro-Angiogenic Activity

Angiogenic properties of MSCs mediated by cell paracrine factors are well described and have been shown to decrease with aging. We evaluated whether pro-angiogenic activity of MSC conditioned media changed after PDGF-BB treatment using the migration endothelial cells (HUVEC) test. HUVEC migrated in the xCELLigence system towards basal media without additives as a negative control, EGM2 containing 2% of FBS as a positive control, and conditioned media (CM) from MSCs or MSCs pretreated with PDGF-BB for 24 h. We observed an increase in HUVEC migration towards CM from MSCs comparable with the positive control (Figure 5A). Moreover, CM from MSCs pretreated with PDGF-BB stimulated the velocity of endothelial cells even more strongly, while PDGF-BB itself decreased it (data not shown). The evaluation of the key pro-angiogenic growth factors, hepatocyte growth factor (HGF) and vascular endothelial growth factor (VEGF), demonstrated a significant increase in the concentration of these factors in CM from PDGF-BB-treated MSCs, compared with CM from control MSCs (Figure 5B).

## 4. Discussion

PDGF-BB is a well-known regulator for stromal cells, such as fibroblasts and MSCs. Produced in high concentrations during the acute phase of tissue damage, it activates stromal cell proliferation and migration [7], and also can modify MSCs’ regulative properties; for example, secretion of extracellular vesicles, changing both protein and RNA composition [22]. PDGFRβ receptor, which has the highest affinity for PDGF-BB dimer [9], is detected on many stromal cells, including MSCs. PDGF-BB is an essential mitogen for many types of stromal cells involved in wound healing as it allows the cell to skip the G1 check point in the cell cycle [23]. During regeneration, PDGF attracts fibroblasts, monocytes, and macrophages into the damage zone, stimulates their proliferation and deposition of the temporary extracellular matrix, and then collagen type I forms the basis of the scar [24]. This process is essential for healing. However, PDGF can apparently trigger the secretion of endogenous factor by macrophages (and possibly other cells) which leads to the formation of an autocrine stimulation loop. Thus, both PDGF-mediated activation of the migration and proliferation of stromal cells into the lesion zone and the switching of the secretory cell phenotype can contribute to fibrogenesis and associated senescence [11].

Since aggressive surrounding in the damaged zone may lead to the induction of cellular senescence, we evaluated whether PDGF-BB could upregulate senescence-related genes within the human MSC transcriptional profile. Groups of upregulated genes were mainly associated with activation of the cell cycle, stimulation of DNA replication, and cell division, indicating that MSCs were actively proliferating after PDGF-BB treatment. This correlated with decreased accumulation of CDK inhibitors, such as p16INK4a, p21Waf1/Cip1, and p53, the markers of senescent cells. Indeed, in validation tests, we did not find any significant elevation in the amount of p21-positive MSCs after PDGF-BB treatment. Many of the proteins coded by the upregulated genes also modulate the reparative response of cells to DNA damage. Such results correlate with the general position on the role of PDGF-BB in the initial stages of response to damage as a crucial mitogen for MSCs.

Importantly, we observed a significant increase in the expression of genes regulating MSC migration, such as PLAU and its receptor PLAUR, MMP-1, and SERPINE1 (PAI-1). These genes are also found to be associated with increased cellular senescence. As was described previously, accelerated aging is associated with cellular senescence and accompanied by marked increases in PAI-1 expression in tissues [25]. PAI-1 is considered as a crucial component of SASP [6]. However, we did not observe any elevation of PAI-1 secretion by PDGF-BB-treated MSCs at the 24 h point. uPAR, as a cell-surface protein, is broadly induced during senescence, and earlier we demonstrated that its surface expression increased in MSCs isolated from adipose tissue of aged patients [26]. Recently, it was shown that uPAR-specific CAR-T cells efficiently ablate senescent cells in vitro and in vivo [27]. On the other hand, both increased levels of the uPA–uPAR axis and its inhibitor PAI-1 paradoxically can stimulate cell migration, probably through activation of different intracellular signaling pathways [28]. Nevertheless, the increased secretion of PAI-1 by PDGF-BB-treated MSCs was not confirmed by ELISA in our validation tests.

Additionally, an increase in the expression of tubulin alpha and beta chain isoform genes, stimulated by PDGF-BB treatment of MSCs, indicates both the activation of proliferation and the possible rearrangement of the cytoskeleton during migration. Additionally, an increase in the expression of the HS3ST3A1 gene (heparin sulfate-glucosamine 3-sulfotransferase), an enzyme involved in the synthesis of heparan sulfate, which plays the role of an adapter in the stimulation of haptotaxis of MSCs through the activation of tyrosine kinase receptors, was detected [29]. These data indicate the migration response of MSCs on the first day of PDGF-BB stimulation and the formation of the supporting mechanisms for homing stimulation.

Among genes downregulated in human MSCs after PDGF-BB treatment, we found THBS2 (thrombospondine-2), which was defined as a protein secreted by senescent cells. It inhibits endothelial migration and prevents cells from senescence escape [30]. However, we received data indicating that conditioned medium from MSC after PDGF-BB treatment had a stimulatory effect on HUVEC directional migration, compared to the initial MSC secretome. This may be due to the upregulation of pro-angiogenic growth factor secretion by MSCs, as well as downregulation of THBS2 expression.

Analysis of clusters of downregulated genes with more than 2-fold changes in expression in PDGF-BB-treated human MSCs showed a decrease in the activity of transcription of pro-inflammatory factors involved in the response to interferons, the innate immune response, and the response to viruses. In addition, there was a strong decrease in IL-6 gene expression, a high level of secretion of which is a characteristic feature of SASP. These effects for IL-6 and MCP-1 were confirmed by ELISA in the experimental validation tests. The obtained data support the acquisition of an anti-inflammatory phenotype for MSCs under PDGF-BB, which is consistent with the idea of the formation of the immunosuppressive MSC phenotype under acute damage. According to the evaluation of widely recognized senescence biomarkers (p21+ and β-gal+) in MSCs, we did not observe any significant difference between control cells and cells after PDGF-BB treatment. ELISA assay showed no significant increase in the level of SASP components, including IL-6, MCP-1, or PAI-1, after MSCs’ incubation with PDGF-BB.

It is important to note that cellular aging is a controversial process, and the development of a senescent phenotype could have some benefits. In a young organism, after damage, senescence of myofibroblasts promotes the inhibition of their proliferation after healing, as well as the involvement of immune cells that eliminate myofibroblasts. The critical contribution of senescent cells in wound healing, injury repair, and tissue regeneration was demonstrated in several studies and related to so-called early senescence [7]. However, if the immune system does not cope with the elimination of senescent cells, they can accumulate in tissues, promoting chronic pro-inflammatory microenvironment, fibrosis, or malignancy of adjacent healthy tissues (late senescence). Switching the normal MSC phenotype to a senescent one can be by the entry of cells into the hostile environment after tissue damage, and PDGF-BB could play a critical role in this process.

Summarizing our data, we conclude that at short periods of exposure to PDGF-BB (up to 24 h), the proliferation of MSCs is activated, and the expression of genes associated with migration increases, as well as the level of secretion of pro-angiogenic growth factors, enabling MSCs to stimulate endothelial migration and angiogenesis in a paracrine manner. However, the analysis of senescent biomarkers in MSCs in such conditions indicates the acquisition of the early senescent phenotype, with benefits for tissue regeneration, but not the signs of late senescence.

## Figures and Tables

**Figure 1 biomedicines-09-01290-f001:**
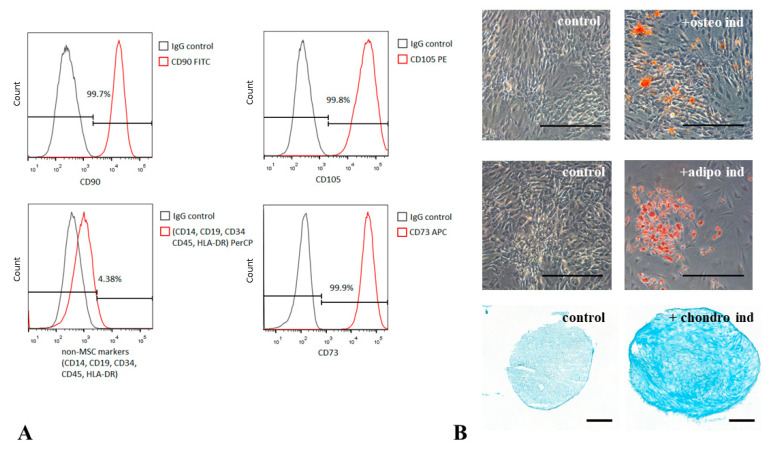
Human adipose-derived MSCs present characteristic features of multipotent mesenchymal stromal cells. (**A**) Evaluation of MSC markers by flow cytometry. More than 99% of MSCs express CD73, CD90, and CD105 on the cell surface. (**B**) MSCs differentiate into osteocytes (Alizarin Red S staining), adipocytes (Oil Red O staining) and chondrocytes (Toluidine Blue staining) under inductive conditions. Representative images, phase contrast microscopy, magnification ×100, scale bar 200 µm.

**Figure 2 biomedicines-09-01290-f002:**
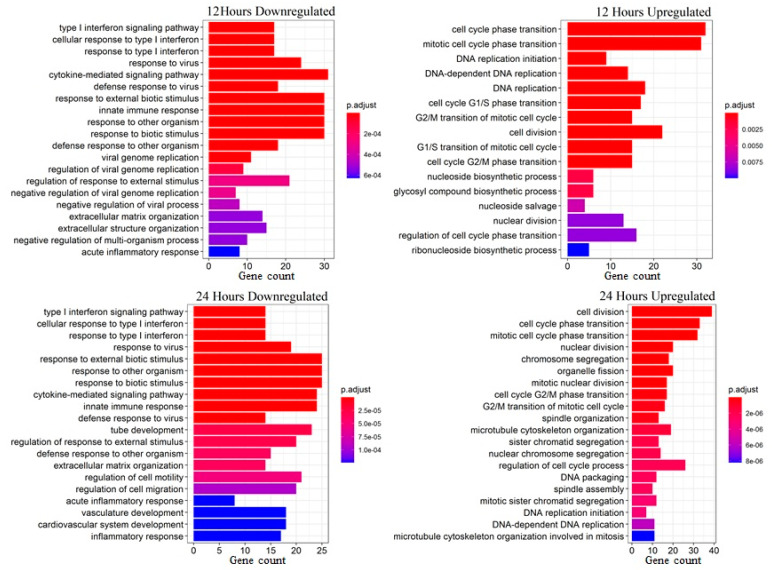
Biological processes associated with differentially expressed genes in human adipose-derived MSCs after 12 and 24 h of PDGF-BB treatment.

**Figure 3 biomedicines-09-01290-f003:**
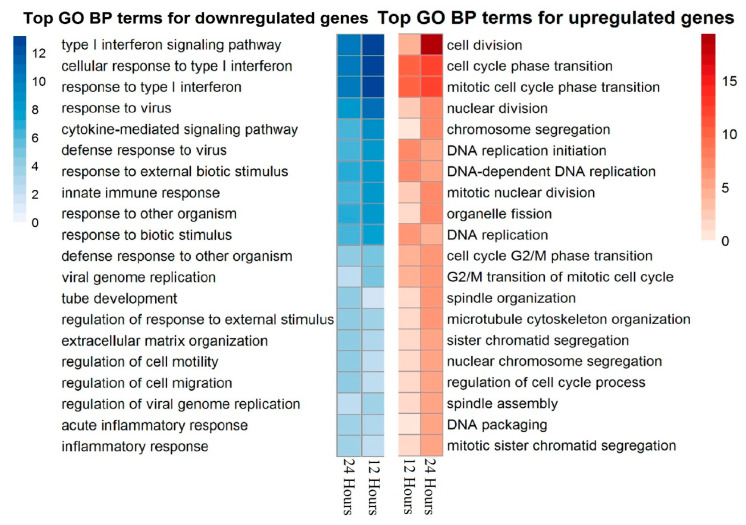
GO BP down/upregulated genes heatmap, 12 and 24 hours after PDGF-BB treatment of human adipose-derived MSCs. The values represent −log10(*p*-value) for enrichment.

**Figure 4 biomedicines-09-01290-f004:**
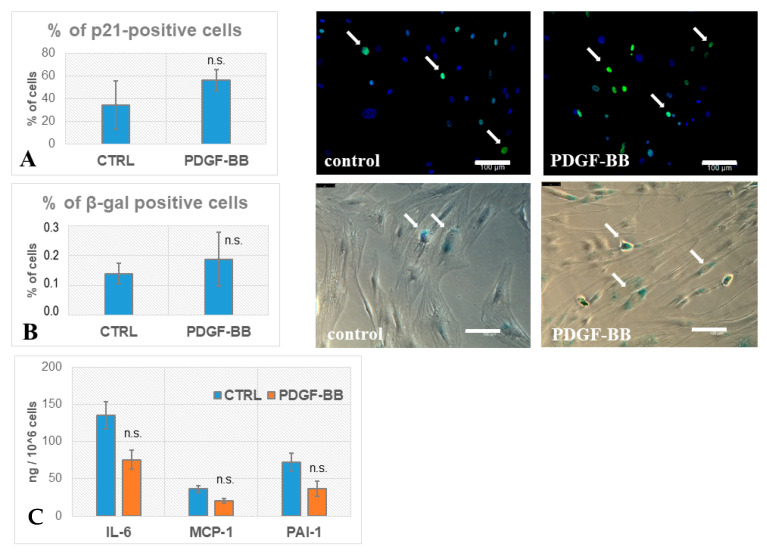
Biomarkers of senescence in PDGF-BB-treated MSCs. (**A**) Immunocytochemical evaluation of p21Waf1/Cip1 in control (CTRL) MSCs and after incubation with PDGF-BB for 24 h. Fluorescent microscopy, green staining—p21Waf1/Cip1, blue staining—nuclei stained with DAPI. The percentage of p21+ cells did not significantly after PDGF-BB treatment. (**B**) Evaluation of β-galactosidase activity (blue staining) in control (CTRL) MSCs and after incubation with PDGF-BB. Phase contrast microscopy, magnification ×200. The percentage of β-gal+ cells did not change significantly after PDGF-BB treatment. (**C**) MSC secretion of IL-6 decreased, while secretion of MCP-1 and PAI-1 was not changed significantly between control (CTRL) and after PDGF-BB treatment. All experiments were performed using cells from three healthy donors in duplicates. Data are presented as mean value of three independent experiments, error bars—standard deviation SD (**B**) or standard error SE (**A**,**C**), n.s.—no significant difference. Scale bar, 100 μm.

**Figure 5 biomedicines-09-01290-f005:**
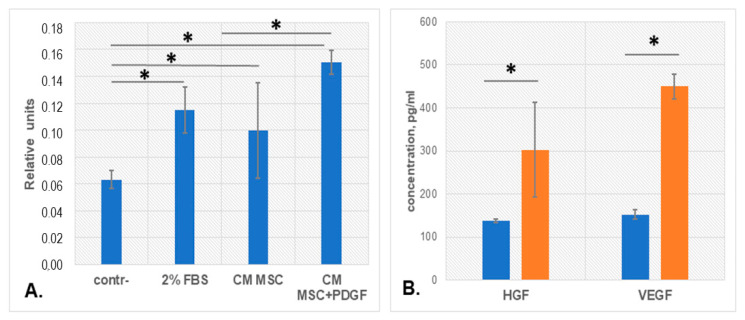
Analysis of MSC pro-angiogenic effects after PDGF-BB treatment. (**A**) The slope (cell index, CI/hour) of HUVEC migrating in the CIM plate of RTCA xCELLigence system towards medium conditioned by MSC (CM MSC) or medium conditioned by MSC after PDGF-BB treatment (CM MSC + PDGF). (**B**) Concentration of growth factors HGF and VEGF in CM MSC (blue) and CM MSC + PDGF (orange). All experiments were performed using cells from three healthy donors in duplicates. Data are presented as mean values of three independent experiments, error bars—standard deviation SD, * *p* < 0.05.

## Data Availability

CHIP files associated with the samples analyzed in this study are deposited in the Gene Expression Omnibus with the accession number GSE152578.

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
