# Peer review of "Platelet-Derived Growth Factor Induces SASP-Associated Gene Expression in Human Multipotent Mesenchymal Stromal Cells but Does Not Promote Cell Senescence"

_biomedicines, 2021, doi:10.3390/biomedicines9101290_

Round 1
Reviewer 1 Report
The article of Olga et al. showed MSC transcriptome changes by PDGF-BB and this study is potentially interesting. Unfortunately, there are serious concerns that prevent the acceptance of the paper.
Major comnments.
- This paper's hypothesis is that PDGF-BB induces MSC senescence. In abstract, they demonstrated that PDGF-BB induced the expression of senescence markers in MSC and did not significantly stimulate their senescence. In result part, PDGF-BB did not induce cellular senescence Figure 4B). I think that this paper is not clear for their conclusion. Although cellular aging is a controversial process, the authors should explain sufficiently for this.
- I think that title of this paper is not appropriate for their data and hypothesis.
Minor comment.
In page 7, PDFG-BB should be changed with PDGF-BB.
Author Response
Responses to Reviewers
On behalf of the authors team I would like to thank Reviewers of our manuscript for their valuable comments, which helped us to improve our manuscript. We have addressed all points raised by reviewers and made changes in the manuscript title, text and figures according to suggestions made by reviewers. All changes are marked up using the “Track Changes” function. I believe that these changes have strengthened our manuscript and made it consistent with the overall quality of Biomedicines.
.Point-by-point response to Reviewer 1
1. This paper's hypothesis is that PDGF-BB induces MSC senescence. In abstract, they demonstrated that PDGF-BB induced the expression of senescence markers in MSC and did not significantly stimulate their senescence. In result part, PDGF-BB did not induce cellular senescence Figure 4B). I think that this paper is not clear for their conclusion. Although cellular aging is a controversial process, the authors should explain sufficiently for this.
Indeed, MSC transcriptome changes in response to PDGG-BB indicated the activation of some senescence-associated gene expression, but we didn’t obtain any evidence of the switching the normal MSC phenotype to senescent one. We interpreted these findings as the acquisition of PDGF-induced early senescent phenotype with activation of MSC proliferation and migration, beneficial for tissue regeneration, rather than the late senescence. However, to make our conclusions more clear we have rewritten the abstract in the revised version of our manuscript.
2. I think that title of this paper is not appropriate for their data and hypothesis.
We agree with the reviewer’s suggestion and have changed the title to “Platelet-derived growth factor induces SASP-associated gene expression in human multipotent mesenchymal stromal cells but does not promote cell senescence”
3. In page 7, PDFG-BB should be changed with PDGF-BB.
Thank you for the comment, the error was corrected.

Reviewer 2 Report
Thank you for this important and very interesting scientific paper on the influence of PDGF-BB on the transcriptome of human MSCs.
There are some linguistic things to improve, both in terms of grammar but also writing style, which would make this article more readable and also more understandable. Here I ask for some improvement.
It is noticeable that the experimental part is sometimes written in a very erratic and superficial manner. That may be enough for the absolute specialist. But we should also take those with us on this beautiful journey who have less background knowledge and have yet to acquire it.
One criticism of the results section is that it should be absolutely limited to the quality of the presentation of the illustrations and the descriptive description of them. Here it is not necessary to generate a context with whatever.
There are some weaknesses in the results section that the editorial office must take into account in the final production.
The discussion is very well written. However, I would like to ask the authors to formulate a conclusion.
Well done, but little improvement should be carried out, to make it to the top.
All suggestions and changes to be made are filled in by track changes, see attached pdf.

Author Response
Responses to Reviewers
On behalf of the authors team I would like to thank Reviewers of our manuscript for their valuable comments, which helped us to improve our manuscript. We have addressed all points raised by reviewers and made changes in the manuscript title, text and figures according to suggestions made by reviewers. All changes are marked up using the “Track Changes” function. I believe that these changes have strengthened our manuscript and made it consistent with the overall quality of Biomedicines.
Point-by-point response to Reviewer 2
We are really grateful for all the revisions and very reasonable comments and suggestions. It is very valuable for us how you thoroughly worked on the text of publication and we do appreciate your contribution.
1. There are some linguistic things to improve, both in terms of grammar but also writing style, which would make this article more readable and also more understandable. Here I ask for some improvement.
We corrected grammar and rewrote recommended parts of the paper to improve perception and make the research understandable to a wider audience.
2. It is noticeable that the experimental part is sometimes written in a very erratic and superficial manner. That may be enough for the absolute specialist. But we should also take those with us on this beautiful journey who have less background knowledge and have yet to acquire it.
One criticism of the results section is that it should be absolutely limited to the quality of the presentation of the illustrations and the descriptive description of them. Here it is not necessary to generate a context with whatever.
We agree with the reviewer suggestions and have carefully elaborated the text of the result section to avoid redundant discussions and imprecise phrasing.
3. The discussion is very well written. However, I would like to ask the authors to formulate a conclusion.
The conclusions were formulated to make the main message of the study more comprehensible.
4. All suggestions and changes to be made are filled in by track changes, see attached pdf.
All the comments were taken into account.
We have indicated the protocol number of the local ethics committee approval and the instructions of which we followed when working with human materials. We have completely changed the 2.6 M&M part according to the suggestions of the reviewer.
However, we would like to explain some points that turned out to be questionable.
First, we realize that full composition of AdvanceSTEM™ Mesenchymal Stem Cell Media, AdvanceSTEM™ Supplement, HyQTase solution, AdvanceSTEM Chondrogenic Differentiation Medium (all by HyClone) would be very useful to know. But these reagents are developed by HyClone to maintain the growth of undifferentiated human mesenchymal stem cells, or to induce chondrogenic differentiation of MSC. The company does not disclose the composition of its products, which is a trade secret, although we know that the supplement contains FBS. That is why we use wording «serum-free AdvanceSTEM™» when we deprive cells from Advance Supplement.
Another point we would like to clarify is timing of different experiments. We analyzed such characteristics of cell senescence in MSCs as p21 expression, β-gal activity, SASP and ability to stimulate endothelial cells migration after 24 hours of exposition to PDGF-BB. According to previous experiments and other researches data this duration is optimal for incubation with PDGF-BB. Although we explored transcriptional changes after 12 and 24 hours of exposition to PDGF-BB to evaluate the dynamics of the process and early transcriptome changes. We assumed the possible «quick-response» changes of genes transcription that we might not have seen in 24 hours.

Round 2
Reviewer 1 Report
I support this work is suitable for publication in Biomedicines.Reviewer 2 Report
Well done, Thx for following my recommendations.